# Methylxanthines Modulate Circadian Period Length Independently of the Action of Phosphodiesterase

Consuelo Olivares-Yañez,[a,b] María P. Alessandri,[a,c] Loreto Salas,[a,c] Luis F. Larrondo[a,c]

[a]ANID-Millennium Science Initiative Program, Millennium Institute for Integrative Biology (iBio), Santiago, Chile
[b]Centro de Biotecnología Vegetal, Facultad de Ciencias de la Vida, Universidad Andrés Bello, Santiago, Chile
[c]Departamento de Genética Molecular y Microbiología, Facultad de Ciencias Biológicas, Pontificia Universidad Católica de Chile, Santiago, Chile

**ABSTRACT**   In *Neurospora crassa*, caffeine and other methylxanthines are known to inhibit phosphodiesterase (PDE) activity, leading to augmented cAMP levels. In this organism, it has also been shown that the addition of these drugs significantly lengthens the circadian period, as seen by conidiation rhythms. Utilizing *in vivo* bioluminescence reporters, pharmacological inhibitors, and cAMP analogs, we revisited the effect of methylxanthines and the role of cAMP signaling in the *Neurospora* clockworks. We observed that caffeine, like all tested methylxanthines, led to significant period lengthening, visualized with both core-clock transcriptional and translational reporters. Remarkably, this phenotype is still observed when phosphodiesterase (PDE) activity is genetically or chemically (via 3-isobutyl-1-methylxanthine) abrogated. Likewise, methylxanthines still exert a period effect in several cAMP signaling pathway mutants, including adenylate cyclase (*cr-1*) and protein kinase A (PKA) (Δ*pkac-1*) mutants, suggesting that these drugs lead to circadian phenotypes through mechanisms different from the canonical PDE-cAMP-PKA signaling axis. Thus, this study highlights the strong impact of methylxanthines on circadian period in *Neurospora*, albeit the exact mechanisms somehow remain elusive.

**IMPORTANCE**   Evidence from diverse organisms show that caffeine causes changes in the circadian clock, causing period lengthening. The fungus *Neurospora crassa* is no exception; here, several methylxanthines such as caffeine, theophylline, and aminophylline cause period lengthening in a concentration-dependent manner. Although methylxanthines are expected to inhibit phosphodiesterase activity, we were able to show by genetic and pharmacological means that these drugs exert their effects through a different mechanism. Moreover, our results indicate that increases in cAMP levels and changes in PKA activity do not impact the circadian period and therefore are not part of underlying effects of methylxanthine. These results set the stage for future analyses dissecting the molecular mechanisms by which these drugs dramatically modify the circadian period.

**KEYWORDS**   methylxanthines, circadian clock, phosphodiesterase, cAMP, protein kinase A

Address correspondence to Luis F. Larrondo, llarrondo@bio.puc.cl.
The authors declare no conflict of interest.

Circadian clocks are present across kingdoms and regulate a variety of biological processes, such as locomotor activity in animals, asexual reproduction in fungi, eclosion in insects, leaf movement and stomatal aperture in plants, etc. (1, 2). In fungi, the circadian clock of the ascomycete *Neurospora crassa* is among the most studied and characterized eukaryotic circadian systems. This oscillator is based on a one-step transcriptional-translational negative feedback loop (TTFL) in which the expression of the *frequency* (*frq*) gene (encoding the negative element) is commanded by the White Collar complex (WCC; positive element), composed of two GATA transcription factors, WC-1 and WC-2. This heterodimeric complex binds to the *clock-box* in the *frq* promoter in the subjective morning activating its transcription. Then, the FRQ protein can inhibit its own expression through FRQ-mediated phosphorylation and inactivation of the WCC (3). This cycle repeats every day, with a periodicity of 22.5 h under

constant conditions. FRQ is an intrinsically disordered protein (IDP) (4–6) and is slowly and progressively phosphorylated throughout the day, modifying its structure and interactors. Several kinases and phosphatases are involved in the regulation of FRQ-WCC interaction and therefore, in determining the length of the circadian period (7, 8).

The circadian clock of *Neurospora* is quite robust to genetic and nutritional perturbations. While several studies have been performed addressing the relevance of transcriptional regulators of the central clock, only few of them have revealed a clear impact in period length (9–13). This contrasts with what has been observed in other circadian systems, where complex transcriptional accessory loops are present and have an impact in period determination (14–16).

In addition to transcriptional regulation, other points of control such as chromatin remodeling, mRNA stability, alternative splicing, and post-translational modifications such as phosphorylation have been described as playing important roles in circadian mechanisms (3, 17–20). The role of the signaling molecule cAMP has also been studied: in several organisms, this second messenger exhibits a circadian expression pattern and seems to play an important role in establishing circadian period length and phase in mammals and insects (21–23).

Nevertheless, in *Neurospora*, it is not clear whether there is such a conserved role for cAMP. In the 1970s, it was reported that the addition of caffeine or other methylxanthines, known inhibitors of phosphodiesterase activity, led to an increase in period length as measured by race tubes, which allow the monitoring of rhythms in asexual spore production, also known as conidial banding (24). Importantly, in *Neurospora*, cAMP regulates different stages of development, conidiation, and glycogen metabolism (25); moreover, protein kinase A (PKA) plays a major role in these processes. However, whether the observed effect of caffeine on banding is at the level of the central oscillator or is solely in the output pathways (development) and what role cAMP plays in this process have been unexplored until now.

In this work, utilizing luciferase-based reporters and genetic and pharmacological perturbations, we evaluated the possible role of phosphodiesterase activity and the relevance of the cAMP-PKA axis in determining the circadian period in *Neurospora*. The evidence compiled here indicates that the effect of methylxanthines occurs independently from their effect on phosphodiesterase activity and that, moreover, elevating cAMP levels or perturbing components of the cAMP-PKA axis does not impact the circadian period in *Neurospora*. However, the data also establish that although the genetic perturbation of adenylate cyclase (AC) has no effect, its pharmacological inhibition leads to subtle period increases, possibly indicating that certain minimum cAMP levels or AC activity are necessary to maintain circadian period within normal range in this fungus.

## RESULTS AND DISCUSSION

**Caffeine and other methylxanthines increase circadian period length.** In the 1970s, Feldman (24) reported that in *Neurospora*, the addition of caffeine lengthened the circadian period in a concentration-dependent manner, as measured by monitoring conidiation rhythms in race tube assays. Nevertheless, for almost 50 years, such observations remained purely phenomenological because no additional data regarding core-clock mechanisms had been obtained for the circadian effects of these drugs in *Neurospora*. Thus, to readdress this, we started by confirming by race tube assays that caffeine and other relevant methylxanthines (theophylline, aminophylline, and theobromine) had significant effects on period length (Fig. 1A). As originally reported, this effect depends on drug levels (Fig. 1B), with up to a ~4.3-h lengthening with a 4-mM concentration of caffeine. To delve into the effects of these drugs on the core-clock, we utilized different luciferase-based *in vivo* reporters reflecting the molecular clockworks (26), confirming period lengthening (Fig. 1C). For these and additional analyses, we focused on only two of these drugs: caffeine and aminophylline. Utilizing transcriptional ($frq_{c\text{-}box}$-*luc*) and translational ($frq^{luc}$) reporters, we confirmed that rhythms in *frq* transcription and FRQ protein levels were indeed lengthened, providing compelling evidence that the effect of these methylxanthines is at the level of core-clock mechanisms and not just due to alterations in conidiation patterns; as in the case of

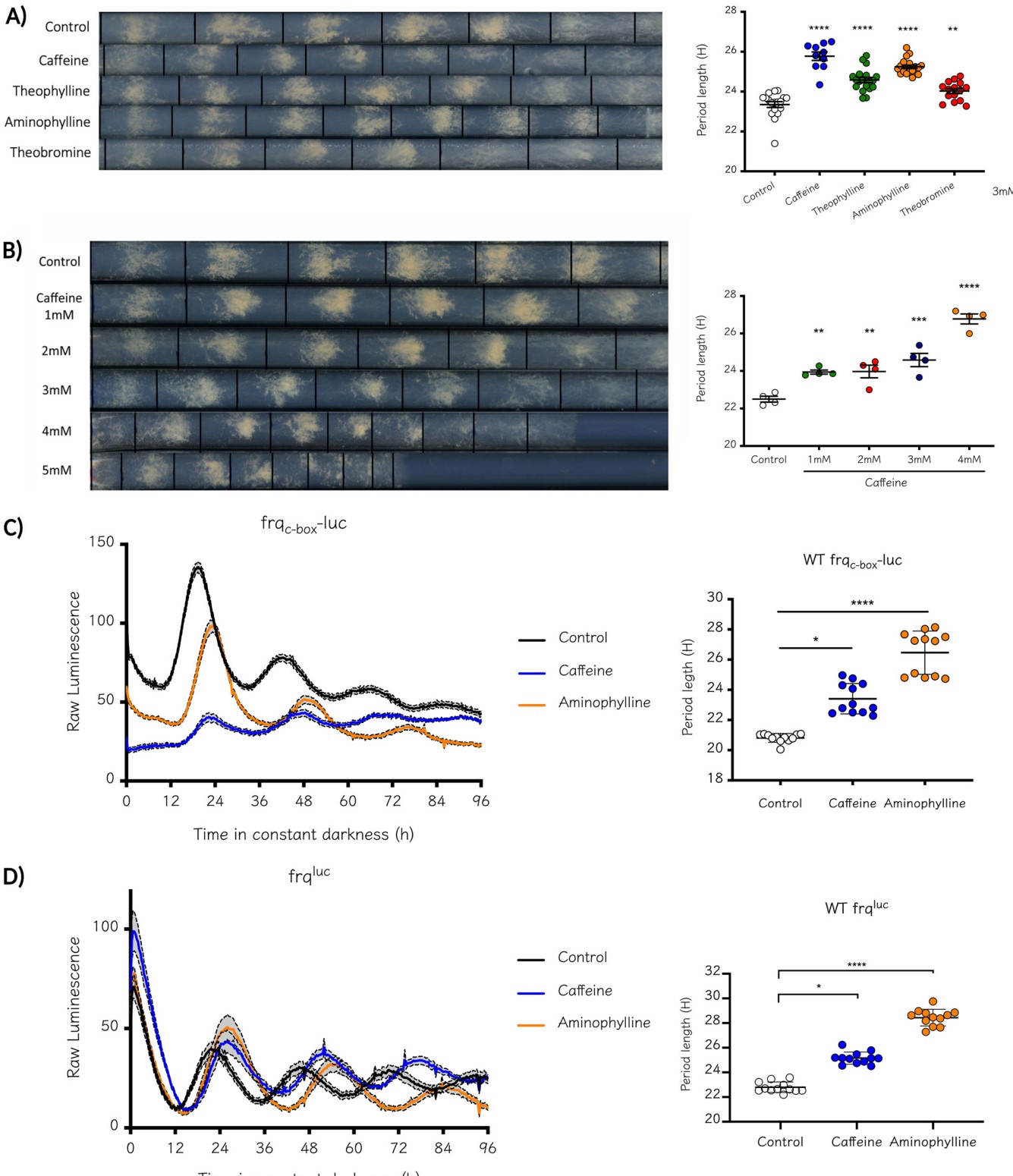

**FIG 1** Caffeine and other methylxanthines increase period length in a concentration-dependent manner in *N. crassa*. (A) The tested methylxanthines increase period length, measured by conidiation in race tubes assays, in *N. crassa*. All drugs were used at a final concentration of 3 mM. Mann–Whitney U test was performed between each treatment and the control. Significance is indicated with an asterisk (*, $P < 0.05$). (B) Effect of caffeine on period length is dose-dependent. Race tubes assays were performed under increasing concentrations from 1 to 4 mM. An increase of up to 4.28 h was observed with a concentration of 4 mM. At high levels of caffeine (5 mM), a severe effect on growth and conidiation was detected and patterns of rhythmic conidiation were hard to observe. One-way analysis of variance (ANOVA) test followed by a Dunnett's multiple-comparison test was performed to evaluate differences between each treatment and the control; significance is indicated with an asterisk (*, $P < 0.005$). (C and D) The increase in period length is also observed at the core-clock level, as evidenced by

*col-1*, where conidial bands respond to metabolic inputs (and fail to be clock-regulated); or Δ*adv-1*, where rhythmic conidiation is absent but FRQ levels oscillate as in the wild type (WT), among other examples (26–29). Importantly, we observed that in the static cultures used for luciferase analyzes, aminophylline consistently yielded greater effects than caffeine on the circadian period; this was not the case in race tubes, where caffeine showed greater effects. The difference in period observed in race tubes and when monitoring luciferase activity in 96-well plates could be a consequence of how the drug may be assimilated by the actively growing fungus (race tubes) versus the fungus in a more stationary state (wells). Notably, a similar phenomenon of different periods when comparing race-tubes and 96-well plates has described in studies of circadian metabolic compensation in this fungus (9, 30, 31). In addition, aminophylline is composed of two methylxanthine molecules which, in a stationary culture, may be constantly metabolized, generating more local methylxanthine units than caffeine.

Importantly, the effects of both drugs on circadian period were also observed when evaluating the expression of a circadian output reporter. For this, we employed a translational fusion between *luc* and *con-10* (*con-10*$^{luc}$). The gene *con-10* is known to be highly expressed during conidiation, exhibiting robust circadian expression (32), and it has been previously utilized to assess the state of the clock (9, 12). As observed (Fig. S2), an increase in period length is detected when caffeine and aminophylline are employed; however, when caffeine was used, we also noted the appearance of a double peak during the first hours, suggesting that caffeine, besides its circadian effect, significantly impacts the expression of this promoter.

**Methylxanthines increase circadian period length even in the absence of their primary target PDE-2.** In different organisms, including *Neurospora*, methylxanthines are known to inhibit phosphodiesterase activity, leading to an increase in cAMP levels (33, 34). In *N. crassa*, there are two phosphodiesterases, PDE-1 and PDE-2/ACON-2, which have low and high affinity for cAMP, respectively. These phosphodiesterases act in a cooperative manner. The absence of PDE-2/ACON-2 significantly impairs growth (Fig. S3) and, indeed, this mutant received its name based on an aconidial phenotype due to a blockage in minor constriction budding (35), suggesting that *acon-2* encodes the main phosphodiesterase in *Neurospora*.

Since methylxanthines influence circadian period (24), one plausible explanation based on the known effect of these drugs on PDEs is that cAMP modulates circadian rhythmicity in *N. crassa*. Thus, to test this hypothesis, we first evaluated the state of the clock in the absence of ACON-2, which would have produced a period defect if this hypothesis was correct. When ACON-2 is missing, cAMP levels should increase because its hydrolysis to AMP is significantly compromised. As shown by two clock reporters, *frq*$_{c-box}$-*luc* and *frq*$^{luc}$ (Fig. 2A), Δ*acon-2* exhibits a functional clock with a period close to a WT strain. Furthermore, when evaluating FRQ protein levels, we confirmed by Western blotting that they were rhythmic in Δ*acon-2* (Fig. 2B), which further validates data indicating rhythms in the historic *acon-2* mutant at its permissive temperature (36). Thus, these results suggest that the absence of ACON-2 does not have a significant impact on core-clock function, particularly relative to period length and amplitude. The lower-affinity phosphodiesterase mutant *(Δpde-1)* has a phenotype comparable to the WT, suggesting that the main phosphodiesterase activity is carried out by ACON-2 (Fig. S3). Unsurprisingly, when period length was evaluated in a Δ*pde-1* strain, its behavior was similar to that of the WT (Fig. S4). We surmised that the activity of PDE-1 might compensate for the absence of ACON-2. To evaluate this hypothesis, we tried, unsuccessfully, to generate a double-mutant strain for phosphodiesterase. Phosphodiesterase activity appears to be crucial for early/critical development in *Neurospora*. Notably, the overt

**FIG 1** Legend (Continued)

transcriptional and translational reporters of *frq* expression, confirming that these drugs affect the central oscillator itself, not just the output. For the luminescence assay, the drugs were used at 3 mM. Period length was calculated using BioDare2 and the FFT-NLLS algorithm. Linear Det. Period length (h) $\pm$ standard deviation (SD). Kruskal-Wallis test followed by Dunn's multiple-comparison test was performed to evaluate differences. *, $P = 0.01$; ****, $P < 0.0001$. (C) *frq*$_{c-box}$-*luc*: Control, 20.82 $\pm$ 0.29; Caffeine, 23.41 $\pm$ 1; Aminophylline, 26.46 $\pm$ 1.44. (D) *frq*$^{luc}$: Control, 22.79 $\pm$ 0.42; Caffeine, 25.16 $\pm$ 0.49; Aminophylline, 28.44 $\pm$ 0.67.

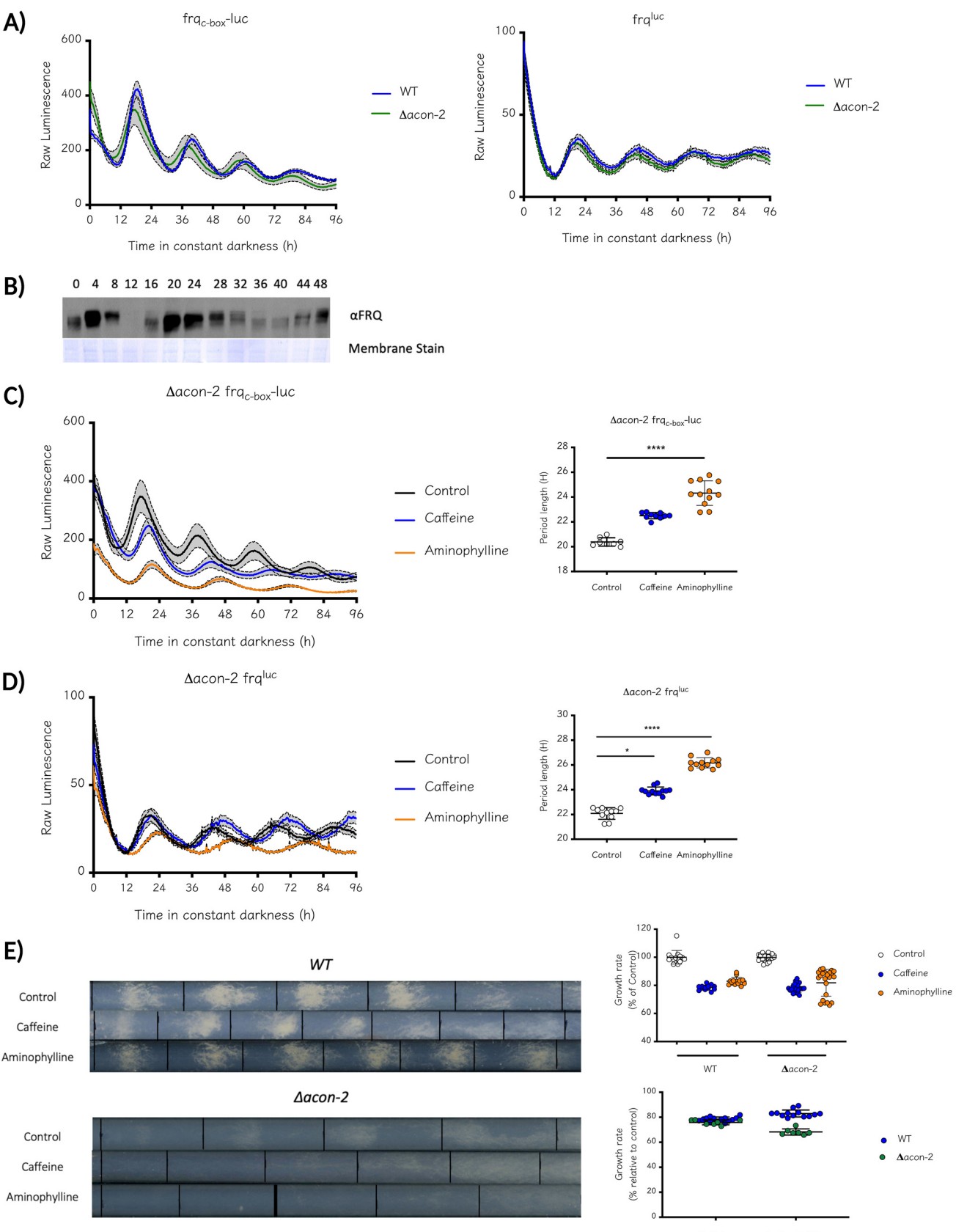

**FIG 2** Phosphodiesterase inhibitors increase circadian period length even in the absence of its primary target PDE-2/ACON-2. (A) The state of the circadian oscillator in the Δ*acon-2* strain was evaluated using transcriptional and translational luciferase reporters, showing a clear circadian behavior with a

developmental phenotypes of Δ*acon-2*, none of which were seen in the *pde-1* mutant, strongly suggest that global PDE activity in the former mutant is strongly compromised.

To further dissect the relationship between PDE, methylxanthines, and circadian period, we also evaluated whether two of these compounds (caffeine and aminophylline) could affect period length in a Δ*acon-2* mutant. As shown in Fig. 2C and D, when monitoring two different core-clock reporters, we confirmed that both drugs increased period even in the absence of what is supposed to be their principal target: ACON-2. An increase of ~2 h was observed with caffeine treatment and of nearly 4 h with aminophylline, differences similar to the effects of these drugs in a WT strain. In addition, the influence of these drugs on Δ*acon-2* was further confirmed with the output reporter *con-10*$^{luc}$ (Fig. S5), corroborating that the lengthening of circadian period is still reflected in the output pathways, independent of the already genetically diminished phosphodiesterase activity.

The addition of caffeine and other drugs also reduces the linear growth of *Neurospora* (Fig. 1A and B). To evaluate whether this effect depends on the inhibition of PDE, WT and Δ*acon-2* strains were grown on race tubes supplemented with 3 mM caffeine or aminophylline (Fig. 2E). In race tubes, circadian conidiation is difficult to assess in the mutant strain because, as its name indicates, it does not produce conidia. However, the addition of methylxanthines clearly reduced growth rates in WT and Δ*acon-2* strains to a similar extent, reinforcing the idea that these drugs also elicit a phosphodiesterase-independent effect in *Neurospora*.

**Circadian period length is not increased by IBMX, a selective inhibitor of phosphodiesterase activity.** As previously indicated, attempts to generate a double *acon-2/pde-2* and *pde-1* mutant through sexual crosses did not yield the desired progeny, suggesting that residual phosphodiesterase activity is crucial at least in the early stages of *Neurospora* development and that PDE-1 likely compensates for essential functions in the absence of ACON-2. It is worth noting, as stated previously, that the strong phenotype displayed by Δ*acon-2* strongly suggests that the latter is responsible for most of the PDE activity in *Neurospora*.

Since one plausible interpretation of the data is that methylxanthines act through an undescribed or noncanonical mechanism(s) different from that of PDE, we evaluated the action of IBMX (3-isobutyl-1-methylxanthine), another phosphodiesterase inhibitor, to further eliminate any role of PDE in regulating clock dynamics. WT reporter strains were incubated with two different concentrations of this drug, and circadian rhythms were observed to continue, with a period slightly shorter than that in the control treatment with the carrier dimethyl sulfoxide (DMSO); this is contrary to what was observed with methylxanthine treatments, where the period was lengthened (Fig. 3). These results indicate that in *Neurospora*, the inhibition of phosphodiesterase activity (genetically or pharmacologically) does not increase circadian period length. In addition, this evidence suggests that the effect of methylxanthines on circadian period cannot rely on the inhibitory role of PDE. When IBMX is used in a Δ*acon-2* strain, the expression levels of both reporters are still affected, as growth is further compromised probably due to the inhibition of residual phosphodiesterase activity (PDE-1) (Fig. S6). However, even in this scenario, no increase in period length was observed; on the contrary, a slight decrease in period length was seen, i.e., for the *frq*$^{luc}$ reporter when 3 mM IBMX was assayed (Fig. S6). It is worth noting that these results differ from what has been reported in other systems, such as mammalian cells and *Drosophila*. In the latter, the addition of IBMX mimics the period length increase observed when chronic doses of caffeine are used (37), while in mammalian cells, the increase in cAMP due to simultaneous AC activation and PDE inhibition dampened the amplitude and definition of the circadian profile (22).

**FIG 2** Legend (Continued)

normal circadian period. Luminescence data were analyzed with BioDare2 and the FFT-NLLS algorithm. Period length (h) ± SD. *frq*$_{c\text{-box}}$-*luc*: WT, 21.38 ± 0.5; Δ*acon-2*, 20.51 ± 0.5. *frq*$^{luc}$: WT, 22.12 ± 0.4; Δ*acon-2*, 22.08 ± 0.5. (B) Western blot of a circadian time course confirms oscillating FRQ levels in this mutant. Methylxanthines increase period length even in a Δ*acon-2* strain as observed by a transcriptional (C) or a translational reporter (D). Both drugs were used at a 3-mM concentration. Luminescence data were analyzed with BioDare2, FFT-NLLS algorithm; results shown as period length (h) ± SD. Δ*acon-2 frq*$_{c\text{-box}}$-*luc*: Control, 20.39 ± 0.34; Caffeine, 22.50 ± 0.25; Aminophylline, 24.32 ± 0.99. Δ*acon-2 frq*$^{luc}$: Control, 22.08 ± 0.47; Caffeine, 23.91 ± 0.31; Aminophylline, 26.17 ± 0.41. (E) WT and Δ*acon-2* were grown in race tubes supplemented with 3 mM caffeine or aminophylline. The addition of methylxanthines reduced growth in both strains (left). Growth rate reduction is similar in WT and Δ*acon-2* strains (right).

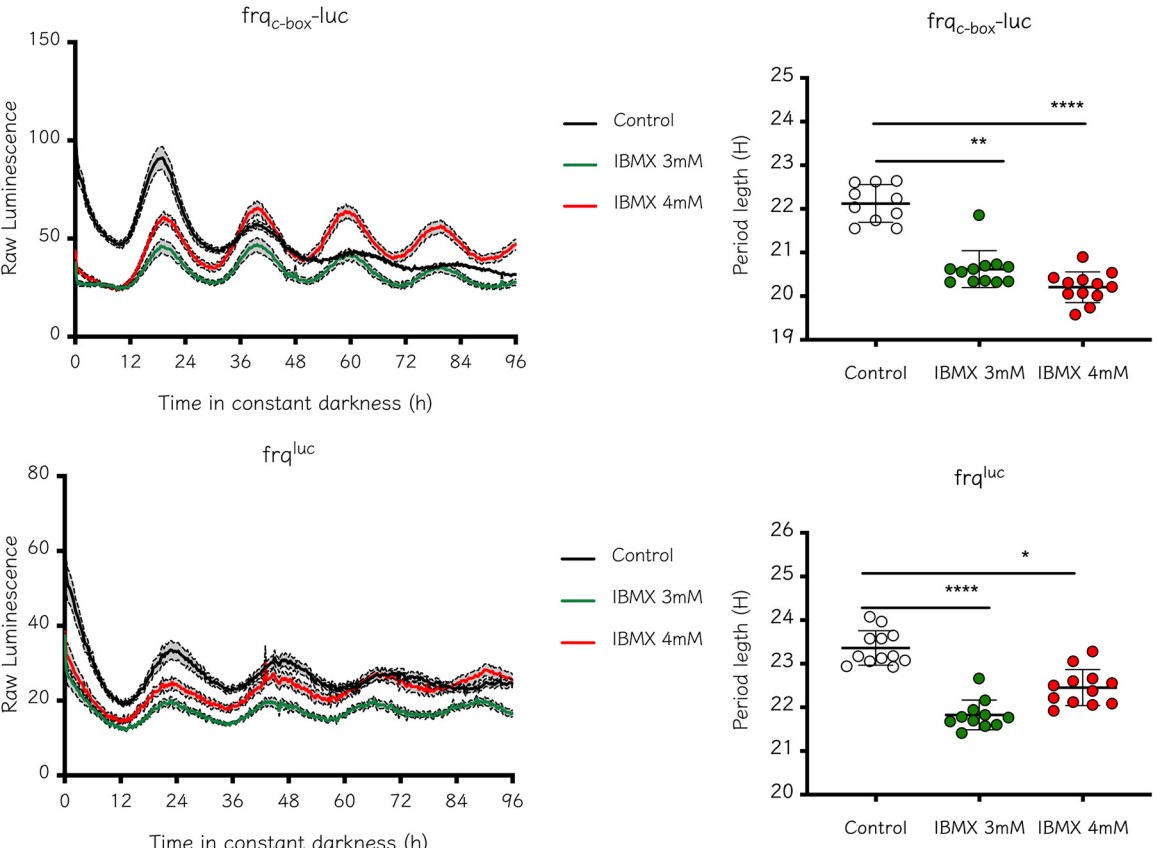

**FIG 3** The addition of IBMX does not increase circadian period length. Circadian period length of WT was evaluated using IBMX (3 or 4 mM), showing a period similar to or shorter than the control. Period was determined using BioDare2 and the FFT-NLLS algorithm. Period length (h) ± SD. Kruskal-Wallis test followed by a Dunn's multiple-comparison test was used to evaluate differences. *, $P = 0.0163$; **, $P = 0.0092$; ****, $P < 0.0001$. $frq_{c\text{-}box}$-*luc*: Control, 22.12 ± 0.43; 3 mM IBMX, 20.62 ± 0.42; 4 mM IBMX, 20.2 ± 0.35. $frq^{luc}$: Control, 23.36 ± 0.39; 3 mM IBMX, 21.82 ± 0.34; 4 mM IBMX, 22.45 ± 0.41.

To further confirm that the effect of methylxanthine was independent of phosphodiesterase activity inhibition, we treated *Neurospora* cultures with a combination of IBMX and aminophylline. While the addition of IBMX does not increase period length, lengthening was observed when aminophylline was added alone or combined with IBMX in the medium (Fig. 4), indicating that even when PDE activity is readily inhibited via IMBX, aminophylline still increases period length. This result reinforces the finding that the inhibition of PDE activity, with the concomitant increase in cAMP levels, is not the mechanism by which methylxanthines increase period length in *Neurospora*, indicating that methylxanthines likely act thought different, as-yet unidentified pathways.

**Broad changes in cAMP levels are not reflected in circadian period.** The result that the inhibition of phosphodiesterase activity with a concomitant increase in cAMP did not affect the circadian period was unexpected. In *Neurospora*, cAMP levels have been shown to fluctuate daily (38); however, the role of this second messenger in *Neurospora* clockworks has not been properly addressed. The importance of cAMP in proper circadian behavior has been described in other organisms. For example, in *Drosophila*, cAMP levels fluctuate daily (21), and mutations that alter the functioning of CREB (cAMP-responsive element-binding protein) alter the circadian clock in this insect (39)—indicating that the cAMP pathway is not only clock-regulated, but also produces feedback to the central oscillator. In mammals, it has been observed that the cAMP signaling cascade has rhythmic behavior, which is necessary for establishing amplitude, phase, and period in the suprachiasmatic nucleus (22).

We have observed that methylxanthines cause an increase in the circadian period which seems to be unrelated to increased cAMP levels, as these drugs still lengthen period

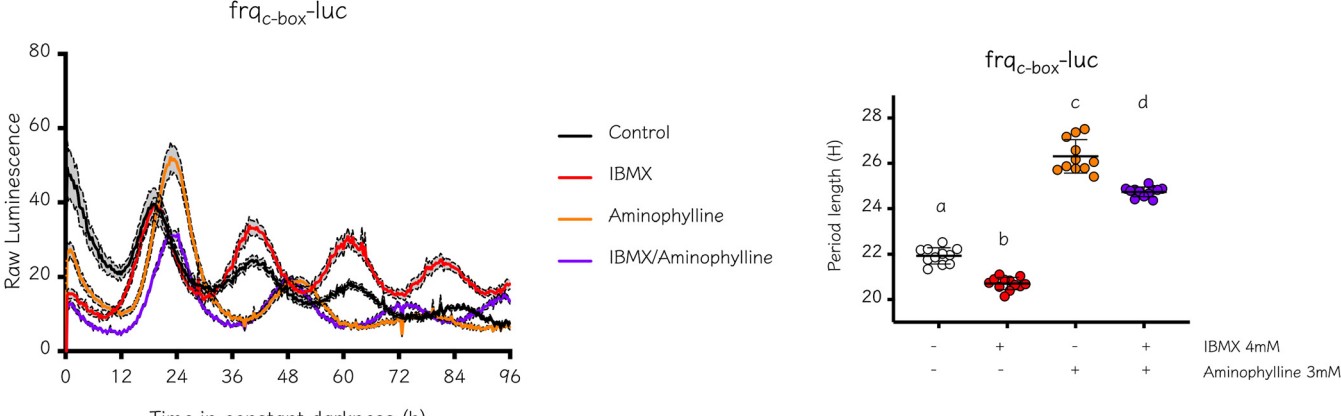

**FIG 4** The increase in period length caused by aminophylline is independent of the inhibition of phosphodiesterase activity. We tested the effect of the addition of a combination of drugs in a WT strain carrying the $frq_{c\text{-}box}$-*luc* reporter. Period determination was performed using BioDare2 and the FFT-NLLS algorithm. Period length (h) $\pm$ SD. One-way ANOVA test followed by Tukey's multiple-comparison test was used to evaluate differences. Different letters indicate significant differences of $P < 0.0001$. Control, 21.93 $\pm$ 0.36; 4 mM IBMX, 20.71 $\pm$ 0.27; 3 mM Aminophylline, 26.31 $\pm$ 0.73; IBMX/Aminophylline, 24.74 $\pm$ 0.22.

even when phosphodiesterase activity is either genetically or chemically inhibited. To further investigate whether cAMP levels impact the *Neurospora* clock, we evaluated whether the external addition of cAMP analogs has any influence on the circadian period. We assessed the effect of two different analogs that have been reported to permeate *Neurospora* cells and are more resistant to phosphodiesterases than cAMP: 8-Br-cAMP and Bt2-cAMP (40, 41). Levels of $frq_{c\text{-}box}$-*luc* and $frq^{luc}$ reporters were evaluated, showing that neither of these drugs significantly modified period length (Fig. S7). These results confirm our previous conclusions: increases in cAMP, either by the inhibition of its degradation (IBMX) or by exogenous addition (cAMP analogs), fail to significantly impact the *Neurospora* circadian clock.

Finally, we evaluated the effect of adenylate cyclase inhibition in core-clock dynamics. AC is the enzyme responsible for generating cAMP from ATP, playing a crucial role in *Neurospora* development, as the null mutant of the corresponding gene is lethal. Therefore, we sought to pharmacologically address AC function with 9-(tetrahydro-2-furyl)-adenine (THFA), a non-competitive AC inhibitor, to reduce cAMP levels (42). Using increasing THFA doses from 0.3 to 1 mM, we observed a subtle increase in period length (Fig. 5), suggesting that certain cAMP levels are indeed required for proper circadian period length. Similar results have been described in mammals and *Ostreococcus tauri*: here, the effect of THFA on the rhythmicity of *mPer1::luciferase* and CCA1-LUC reporters was evaluated, with increases in period observed upon treatment with this drug (22, 43). In *Neurospora*, a small change in phases is observed with THFA treatment. Nevertheless, this seems to be related to the increase in period length, because when we estimated the phase using circadian time (CT), the values were very similar to the control.

Importantly, the *Neurospora* mutant *cr-1* exhibits compromised AC activity (41, 44) leading to extremely low cAMP levels (45). Therefore, we evaluated the state of the oscillator in this mutant, observing that despite its very low levels of intracellular cAMP, circadian rhythms still occur (Fig. S8A). Notably, the expression of the output clock reporter *con-10*$^{luc}$ is clearly altered in this strain (Fig. S8B), which is to be expected because *con-10* is a gene downstream of the cAMP-PKA axis (46). In agreement with our prior conclusions, the addition of caffeine to *cr-1* still led to an increase in period length (Fig. S8C).

Still, to make sure that intracellular cAMP levels were behaving as expected, we quantified cAMP levels by an enzyme-linked immunosorbent assay (ELISA, Fig. S9). As a control, we measured the levels of cAMP in the mutant strain *cr-1*. As previously described, we confirmed very low levels of cAMP in this mutant (41). After treatment with caffeine (3 mM), an increase in cAMP levels was observed in the WT strain, while higher levels of cAMP are observed in Δ*acon-2* compared to the WT; this is expected, considering that in the absence of the major phosphodiesterase, cAMP degradation is impaired. Importantly, our quantification also indicated that the addition of caffeine to Δ*acon-2* does not further increase cAMP

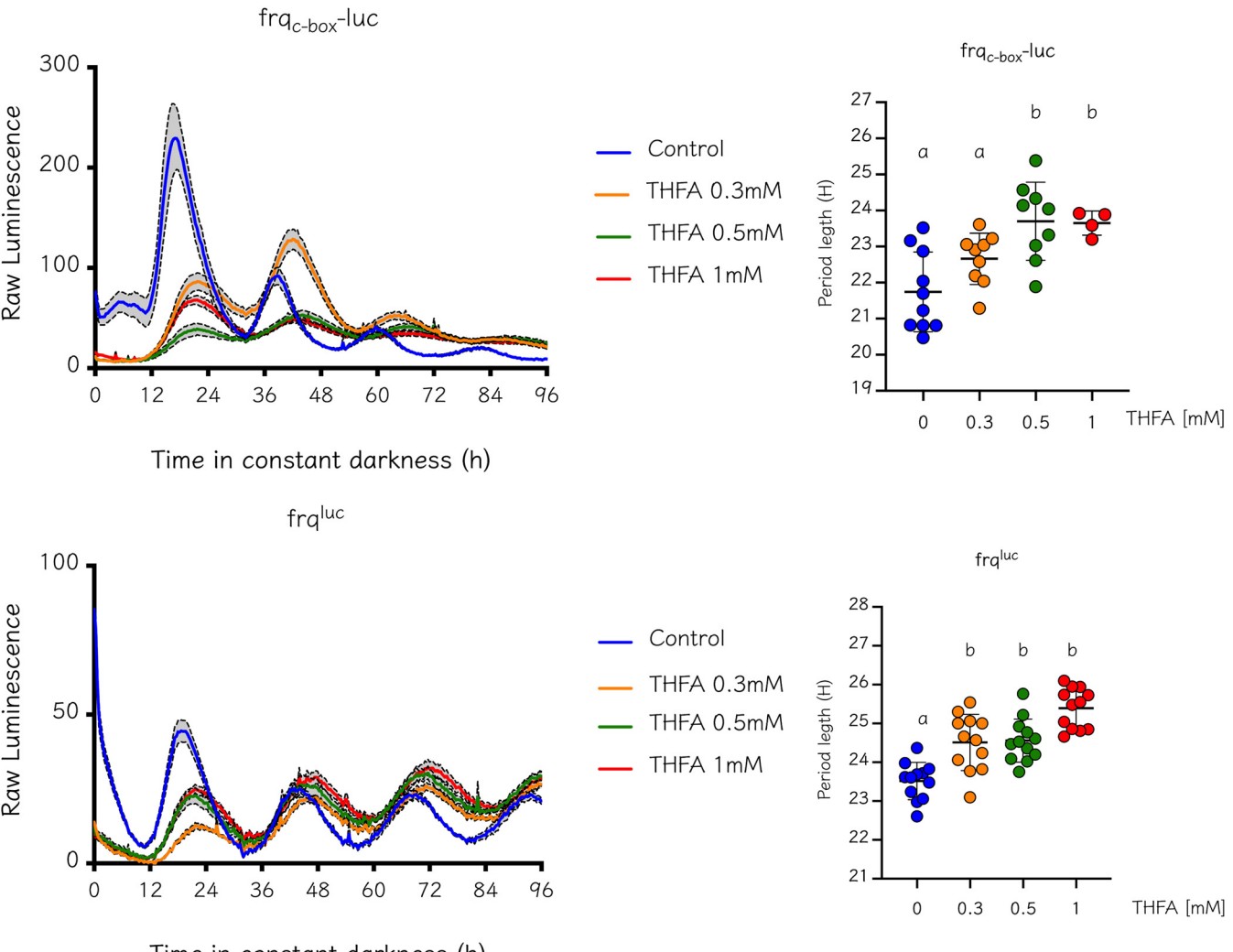

**FIG 5** Pharmacological inhibition of AC leads to an increase in circadian period length. To evaluate whether a decrease in cAMP levels affects period length, we blocked cAMP synthesis using THFA, a non-competitive inhibitor of adenylate cyclase. Increasing concentrations of this drug were added, showing a dose-dependent increase in period length for both clock reporters. Prior to luciferase measurement, plates were inoculated and incubated for 24 h in LL conditions. Period determination was performed using BioDare2 and the FFT-NLLS algorithm. Period (h) $\pm$ SD. Kruskal-Wallis test followed by Dunn's multiple-comparison test was used to evaluate differences. A dose-dependent increase in circadian period length was observed for both reporters. *frq*$_{c\text{-box}}$-*luc*: Control, 21.74 $\pm$ 1.1; 0.3 mM THFA, 22.66 $\pm$ 0.71; 0.5 mM THFA, 23.59 $\pm$ 1.1; 1 mM THFA, 23.65 $\pm$ 0.33. *frq*$^{luc}$: Control, 23.5 $\pm$ 0.48; 0.3 mM THFA, 24.51 $\pm$ 0.72; 0.5 mM THFA, 24.56 $\pm$ 0.56; 1 mM THFA, 25.39 $\pm$ 0.52.

levels, suggesting that caffeine is indeed able to increase cAMP levels by inhibiting PDE activity and that most of the PDE activity is due to ACON-2. Thus, the fact that caffeine in $\Delta acon$-2 does not further increase cAMP content, but still affects period length, further supports our interpretation of a cAMP-independent mechanism (Fig. 2). Along the same lines, we observed a marked increase in intracellular cAMP levels when the WT strain was grown with the cAMP analog 8-Br-cAMP (Fig. S9); this emphasizes that this analog is capable of permeating the *Neurospora* cell membrane and that, despite its high intracellular levels, it does not affect period length (Fig. S7).

Thus, our results indicate that, with the exception of the experiments with THFA, cAMP levels have no obvious impact on circadian period. On the one hand, methylxanthines are expected to elevate cAMP levels, which should underly period lengthening. However, when we genetically or pharmacologically targeted PDE activity, period length was not increased (Fig. 2 and 3), suggesting that high levels of cAMP (as confirmed by measurements, see Fig. S9) are likely not causative of the circadian effect of methylxanthines. On the other hand, when we genetically compromised AC activity (leading to low cAMP levels), period was unaffected. However, when we pharmacologically blocked AC activity, increased

period length was observed (Fig. 5); this may suggest unforeseen aspects of AC on the clockworks or alternatively indicate off-target effect of THFA.

**Protein kinase A activity is not involved in modulating the clockworks.** Importantly, in *Neurospora*, PKA is a relevant downstream effector of cAMP. With this in mind and considering the critical role of kinases in determining the circadian period in different organisms (47, 48) we considered evaluating the potential role of the *Neurospora* cAMP-PKA axis in period determination. In *Neurospora*, the cAMP-PKA axis is composed of one adenylate cyclase, CR-1 (NCU08377); one PKA regulatory subunit, MCB (NCU01166); and two PKA catalytic subunits, PKAC-1 (NCU06240) and PKAC-2 (NCU00682) (46). Of the two protein kinase subunits, PKAC-1 is accountable for most of the cellular PKA activity (49).

In brief, when cAMP levels are low, the regulatory subunit MBC is bound to the catalytic subunit; therefore, the complex remains inactive. When cAMP levels increase, cAMP binds to MBC, promoting a conformational change that separates it from the catalytic subunit, allowing the activation of this complex (46). Thus, an increase in cAMP levels will lead to augmented PKA activity, whereas low cAMP levels produce the opposite. Therefore, to establish whether the variation in cAMP levels could modify period length through the modulation of PKA activity, we first evaluated the state of the circadian clock in the absence of PKAC-1. The latter component had been originally reported as being essential for clock function, as assayed by conventional methods such as race tubes and Western blotting. Importantly, *pkac-1* mutants exhibit impaired morphological development, displaying colonial growth and hyperconidiation (49); therefore, circadian conidiation in race tubes is challenging to assess (as shown in Fig. 6B). Nevertheless, the use of *in vivo* luminescence reporters allows us to accurately assess the state of the central core-clock even in strains that may be difficult to approach with conventional assays due to poor growth. When analyzing $frq_{c-box}$-*luc* or $frq^{luc}$ in the Δ*pkac-1* mutant, we confirmed that the absence of PKAC-1 does not alter circadian function (Fig. 6B), limiting the essential role first assigned to this kinase in the *Neurospora* clock (50, 51). Thus, we can observe clear and robust rhythms in both reporter strains, which could also be confirmed when tracking FRQ levels in cultures coming from solid medium conditions (Fig. 6C). These results showed that under our experimental conditions, PKAC-1 is not necessary for the generation of circadian rhythms in *Neurospora*, echoing results that were recently reported for this kinase (31, 52). Similarly, we evaluated the state of the central oscillator in the Δ*pkac-2* mutant strain (Fig. S10A), obtaining a similar profile to a WT strain, in agreement with the notion that most of the cellular PKA catalytic activity is carried out by PKAC-1 (49). In addition, to rule out any role of PKA activity in the clockworks such that the rhythms observed in Δ*pkac-1* might be fueled by residual activity from PKAC-2, we obtained a double mutant of both kinases through sexual crosses. Evaluation of the Δ*pkac-1/pkac-2* mutants confirmed circadian rhythms similar to that of Δ*pkac-1*, with a period close to that of the WT strain, clearly showing that PKA activity has no perceptible effect on the clockworks, at least under standard growth conditions (Fig. S10B).

Furthermore, when we analyzed the expression of the CON-10 output reporter in Δ*pkac-1*, we also observed clear circadian expression of this gene (Fig. S11A), albeit with a slightly shorter period than that of the WT strain (Fig. S11B). As noted, CON-10 levels are considerably higher in comparison to those in WT or Δ*acon-2* (Fig. S11C), confirming that the cAMP-PKA pathway regulates *con-10* expression.

Thus, these results indicate that any subtle effect cAMP levels may have on the clock (as suggested from the THFA experiments) are likely not mediated by PKA; in fact, preliminary results have shown that the increase in period length generated by the addition of THFA persists in the *pkac-1* mutant strain.

**Methylxanthines increase period length in the absence of PKAC-1.** After obtaining evidence of clock functionality in the absence of PKAC-1, we evaluated whether the circadian effect of methylxanthines is mediated by PKA and confirmed that, in a Δ*pkac-1* strain, an increase in period length was observed when caffeine or aminophylline was added (Fig. 6D). We also tested the effect of caffeine on Δ*pkac-1/pkac-2*, finding that it still led to period lengthening (Fig. S10C). These results validate the conclusion that the cAMP-PKA axis does not have a preponderant role in period determination in *Neurospora* and that methylxanthines lengthen period through an as-yet undefined mechanism.

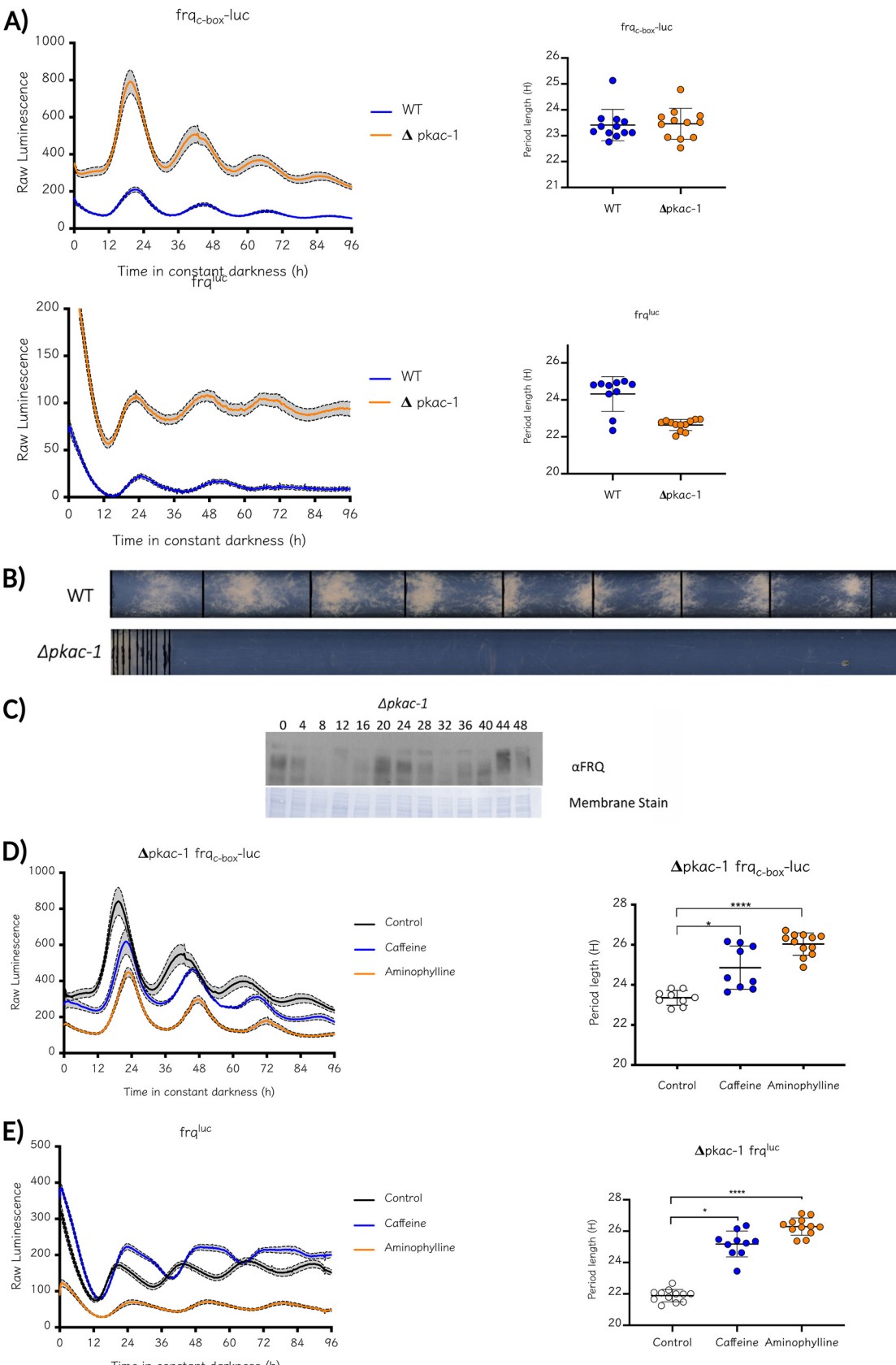

**FIG 6** Circadian rhythms remain normal in the absence of protein kinase A. The Δ*pkac-1* mutant exhibits severe growth defects and circadian conidiation cannot be observed in race tubes (A) The *in vivo* transcriptional and translational reporters *frq*$_{c-box}$*-luc*

**Conclusion.** In summary, our results confirm that in *Neurospora*, the cAMP signaling pathway does not have a predominant role in controlling circadian period length, unlike what has been described in other circadian systems. This is particularly evident when observing normal periods in strains with high cAMP levels (Fig. S9), such as WT grown with 8-Br-cAMP, the mutant Δ*acon-2*, or strains displaying low cAMP levels, such as the *cr-1* mutant. Similarly, the use of the phosphodiesterase inhibitor IBMX did not increase period length and instead caused a slight decrease in period length, suggesting some side effects of this drug beyond its classic effects on PDE. Notably, the effect of addition of methylxanthines on period length still occurred in the *acon*-2 mutant and was also observed when a combination of IBMX and methylxanthine was used (Fig. 4).

Literature from different model systems indicates that alteration of cAMP levels affects clock function. Therefore, aspects of this signaling pathway are considered "part of the clock." However, our results indicate that in *Neurospora*, this phenomenon seems to be different, and that the cAMP-PKA signaling axis does not play a fundamental role in the central oscillator. Moreover, in our hands, the only condition that led to a period change (increase) when tampering with this axis was when we utilized the AC inhibitor THFA, yet this outcome was not present in the AC mutant *cr-1*; this result could indicate off-target effects of this drug besides its known effects on AC.

Taking these results together, our data indicate that the effect of caffeine and other methylxanthines on circadian rhythm implies a mechanism different from PDE inhibition and the concomitant changes in cAMP. What these mechanisms are will be the subject of future research. In addition, our results also reinforce the idea that despite initial reports implicating PKA in the clockworks, this kinase is neither essential for the core-clock mechanisms nor relevant (under the tested conditions) for period determination.

## MATERIALS AND METHODS

**Strains, culture conditions, and race tubes assays.** The strains FGSC 2489 (*wt*), FGSC 11513 (Δ*pkac-1*, accession no. NCU06240), FGSC 11433 (Δ*pkac-2*, NCU00682), FGSC 11431 (Δ*acon-2*, NCU00478), FGSC 11429 (Δ*pde-1*, NCU00237), and FGSC 4008 (*cr-1*, NCU08377) were used in this study. For race tube and luciferase assays using the reporter *frq*$_{c-box}$-*luc*, these strains were crossed with a *his-3::frq*$_{c-box}$-*luc*, *ras-1*$^{bd}$ reporter strain. Strains were grown in constant light (LL) at 25°C in slants with Vogel medium (1× Vogel's, 2% sucrose, 1.5% agar [pH 5.8]). For the analysis of *in vivo* levels of *frq* or *con-10* expression, strains containing translational fusion of these open reading frames (ORFs) with luciferase were crossed with selected mutants obtained from the FGSC. In all cases, to properly select the mutant strains, DNA was extracted and the strains were genotyped through PCR utilizing primers which amplify the ORF of each gene and the KO cassette (Table 1). Although the parental strain FGSC 11513 (Δ*pkac-1*) is a heterokaryon, after sexual crossing, homokaryotic strains of Δ*pkac-1* were obtained and genotyped as such through PCR (Fig. S1). Δ*pkac-1/pkac-2* double mutants were obtained through sexual crossing by selecting them at higher hygromycin concentrations (600 μg/mL) and were genotyped through PCR (Fig. S1).

For race tube assays, 18 mL medium containing 1× Vogel's salts, 0.03% glucose, 0.05% arginine, 50 ng/mL biotin, and 1.5% agar was used per race tube. After autoclaving, the medium was supplemented with the different drugs as indicated. Due to solubilization issues, theobromine was directly added as powder to the melted medium to reach its final concentration. To synchronize the clock, the strains were inoculated and kept in LL at 25°C for 24 h before transferring them to constant darkness (DD). Under safe red light, the growth front was marked every 24 h. Period determination was performed using ChronOSX v2.1 software (53). At least four biological replicates per condition were analyzed.

**Luciferase reporter assays.** Unless specified otherwise in the figure legends, all *Neurospora* strains were inoculated in 96-well plates and grown at 25°C in a 12:12-h light-dark (LD) cycle for 72 h before transfer into DD for luciferase monitoring. The luciferase analyses were performed as previously described (26). Briefly, 96-well plates were inoculated with LNN-CCD medium (1× Vogel's salts, 0.03% glucose, 0.05% arginine, 50 ng/mL biotin, 1.5% agar), supplemented with luciferin at a final concentration of 25 μM. Luciferase signals

**FIG 6** Legend (Continued)

and *frq*$^{luc}$, respectively, reveal the clear presence of circadian rhythms in luciferase in the absence of PKAC-1, with a circadian period close to that of the WT, even when circadian conidiation is impaired (B). These rhythms are also observed at the protein level through Western blot assays (C). Period determination of luciferase reporters was performed using BioDare2 and the FFT-NLLS algorithm. A Mann–Whitney U test was performed between each treatment and the control. No significant difference was found. Period length (h) ± SD. *frq*$_{c-box}$-*luc*: WT, 23.41 ± 0.60; Δ*pkac-1*, 23.41 ± 0.60. *frq*$^{luc}$: WT, 24.32 ± 0.94; Δ*pkac-1*, 22.64 ± 0.3. (D) The addition of methylxanthines increases the period length as shown in both reporter strains, even in the absence of PKAC-1. Period determination was performed using BioDare2 and the FFT-NLLS algorithm. Period length (h) ± SD. Kruskal-Wallis test followed by Dunn's multiple-comparison test was used to evaluate differences. *, $P < 0.05$; ****, $P < 0.0001$. Δ*pkac-1 frq*$_{c-box}$-*luc*: Control, 23.35 ± 0.36; 3 mM Caffeine, 24.85 ± 1.08; 3 mM Aminophylline, 26.03 ± 0.58. Δ*pkac-1 frq*$^{luc}$: Control, 21.89 ± 0.39; 3 mM Caffeine, 25.18 ± 0.82; 3 mM Aminophylline, 26.28 ± 0.55.

**TABLE 1** Sets of primers used for strain genotypification[a]

| Primer set | Usage | Orientation | Sequence (5′→3′) |
| --- | --- | --- | --- |
| LC222 | To amplify the *pkac-1* ORF | FW | ATGCCTCTGCCTAGCCTCGG |
| LC223 | | RC | AGGTGGCAGGTTGGTCGTAT |
| oL116 | To amplify the KO cassette in Δ*pkac-1* | FW | GCCATGTAGTGTATTGACCG |
| oL2458 | | RC | AACTTTCAGAACCAGGATTC |
| LC328 | To amplify the *pkac-2* ORF | FW | CCGTCGCCTCGTCACACCCA |
| LC329 | | RC | GCCGGTTACATCCTGGAGGA |
| oL116 | To amplify the KO cassette in Δ*pkac-2* | FW | GCCATGTAGTGTATTGACCG |
| oL2457 | | RC | GCTTTTGCAATGTGTGTACG |
| oL3027 | To amplify the *acon-2* ORF | FW | TCGCAGTGATTCAAGACAGG |
| oL3028 | | RC | GATCGTTTTCTGCCCTTACG |
| oL116 | To amplify the KO cassette in Δ*acon-2* | FW | GCCATGTAGTGTATTGACCG |
| oL2640 | | RC | CCAATGACAACCGTCTCTGG |
| oL3029 | To amplify the *pde-1* ORF | FW | CAACGCCGATGAGTGTAATG |
| oL3030 | | RC | TAGCTGCCGATATGCTGTTG |
| oL116 | To amplify the KO cassette in Δ*pde-1* | FW | GCCATGTAGTGTATTGACCG |
| oL2461 | | RC | CAGAGGCAGCGAGTAGAAAA |
| oL810 | To amplify the *ras-1*[WT] allele | FW | CCTGATTTCGCGGACGAGATCGTA |
| oL811 | | RC | GCGCGAGCAGTACATGCGGAC |
| oL810 | To amplify the *ras-1*[bd] allele | FW | CCTGATTTCGCGGACGAGATCGTA |
| oL812 | | RC | TGCGCGAGCAGTACATGCGAAT |
| oL808 | To amplify a 1,000-bp fragment used as control | FW | CCTTCAATATCATCTTCTGTCGAGTCTAGAAAACACCAGCGGAGTTGCAA |
| oL809 | | RC | GTAACGCCAGGGTTTTCCCAGTCACGACGAGAAAAAGACAAAGGTCAAG |

[a]ORF, open reading frame; FW, forward; RC, reverse; KO, knockout.

were acquired with a PIXIS 1024B camera (Princeton Instruments) and quantified with a customized macro developed for Image J software; luminescent signals were acquired for 10 min, 3 times per hour, over the indicated times in DD. For the reporter *con-10*[luc], the luminescent signal was captured for 5 min 3 times per hour. Most of the used drugs were freshly prepared and sterilized by filtration (0.22-$\mu$M filter). Caffeine, theophylline, and aminophylline stocks were prepared at 100 mM (cat no. C0750, T1633, A1755; Sigma), 8-Br-cAMP at 25 mM (cat no. B7880, Sigma) and Bt2-cAMP at 40 mM (cat no. D0260), all in water; and IBMX 500 mM (cat no. I5879, Sigma) and THFA 50 mM (cat no. SQ22536, R&D Systems) in DMSO. Theobromine was added directly to the culture medium. Period determination analyses were performed using BioDare2 software (biodare2.ed.ac.uk) and the FFT-NLLS (fast Fourier transform non-linear least squares) algorithm (54). Linear detrending was conducted before period estimation. At least four biological samples with three technical replicates were used for period quantification.

**Time-course experiments.** To assess FRQ protein levels, time-course experiments were performed in DD conditions, on solid medium, as previously described (9). Briefly, the strains were pre-grown in petri plates with 25 mL liquid medium containing 1× Vogel's salts, 2% sucrose, 0.2% Tween 80, 0.5% arginine, and 50 ng/mL biotin for 48 h in LL. From here, 5-mm tissue pads were cut and inoculated in cellophane-covered petri plates with 30 mL LNN-CCD solid agar medium. For each time course, 13 plates were inoculated and incubated in LL for 4 h, after which plates were transferred individually every 4 h to DD conditions. After 48 h, cultures were harvested under a safe red light, freeze-dried, and ground in liquid nitrogen, and proteins were extracted.

**Protein extraction and Western blotting.** Protein lysates and Western blot analyses were performed as described previously (55), with slight modifications. Protein lysates were prepared with DTT, PSMF and Halt Protease Inhibitor Cocktail (cat no. 78438, Thermo Fisher Scientific). Protein concentration was quantified by Bradford assays. For Western blot analysis, 100 $\mu$g of total protein was loaded per lane. Polyclonal Anti-FRQ antibody was diluted 1:250 in 1× PBS (phosphate-buffered saline) + 0.03% Tween + 1% bovine serum albumin. Anti-rabbit-HRP (horseradish peroxidase) secondary antibody was used at 1:5,000 dilution (cat no. 1706515, Bio-Rad) in 1× PBS + 0.03% Tween. SuperSignal West Femto Maximum Sensitivity Substrate was used for signal development (cat no. 34096 Thermo Fisher Scientific). Membranes were stained with Blue-Coomassie R-250 (cat no. 1610400, Bio-Rad) and used as a loading control.

**Measurement of cAMP levels.** The strains were pre-grown in petri plates with 25 mL liquid medium as described previously in the "Time-course experiments" section. After 48 h, 5-mm tissue pads were cut and inoculated in cellophane-covered petri plates with 25 mL LNN-CCD solid agar medium supplemented with the indicated drugs. The drugs were freshly prepared and sterilized by filtration. Caffeine was prepared at 100 mM, 8-Br-cAMP at 25 mM in water, and IBMX 500 mM in DMSO. After 48 h of growth under the LL condition, tissue was harvested, freeze-dried, and lyophilized. The dry weight of each sample was measured. Intracellular cAMP levels were determined using TCA extraction and a commercial ELISA kit (cat no. 581001, Cayman Chemical, Ann Arbor, MI, USA) according to the manufacturer's recommendations.

## SUPPLEMENTAL MATERIAL

Supplemental material is available online only.
**SUPPLEMENTAL FILE 1**, DOCX file, 7 MB.

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
