## [Reviewer comments · Microbiology Spectrum]

Microbiology Spectrum

Methylxanthines modulate circadian period length independently of the action of phosphodiesterase

Consuelo Olivares-Yañez, Maria Alessandri, Loreto Salas, and Luis Larrondo

Corresponding Author(s): Luis Larrondo, Pontificia Universidad Catolica de Chile

Review Timeline:

Submission Date:	September 13, 2022
Editorial Decision:	December 3, 2022
Revision Received:	April 30, 2023
Accepted:	May 22, 2023

Editor: Rebecca Shapiro

Reviewer(s): The reviewers have opted to remain anonymous.

Transaction Report:

DOI: <https://doi.org/10.1128/spectrum.03727-22>

December 3, 2022

Dr. Luis F Larrondo
Pontificia Universidad Catolica de Chile
Genetica Molecular y Microbiologia
Alameda 340
Alameda 340
Santiago, RM
Chile

Re: Spectrum03727-22 (Methylxanthines modulate circadian period length independently of the action of phosphodiesterase)

Dear Dr. Luis F Larrondo:

Thank you for submitting your manuscript to Microbiology Spectrum. Two reviewers have assessed the manuscript, and identified some modifications requested before consideration for publication. When submitting the revised version of your paper, please provide (1) point-by-point responses to the issues raised by the reviewers as file type "Response to Reviewers," not in your cover letter, and (2) a PDF file that indicates the changes from the original submission (by highlighting or underlining the changes) as file type "Marked Up Manuscript - For Review Only". Please use this link to submit your revised manuscript - we strongly recommend that you submit your paper within the next 60 days or reach out to me. Detailed instructions on submitting your revised paper are below.

Link Not Available

Sincerely,

Rebecca Shapiro

Journals Department
Reviewer comments:

Reviewer #1 (Comments for the Author):

Olivares-Yañez and colleagues revisit classical data from 1975 (Feldman JF, Science, PMID: 173018) showing that *Neurospora crassa* circadian period lengthens significantly (~2-4) hours in the presence of theophylline, aminophylline, and caffeine. Olivares-Yañez presents a careful analysis using multiple circadian reporter strains to show that, surprisingly, the period-lengthening effect of these methylxanthine drugs is largely independent of the phosphodiesterase/cAMP/PKA signaling pathway.

This work presents a strong genetic argument that inhibition of phosphodiesterase activity is not the causal mechanism with which methylxanthine drugs lengthen period (Fig 2, Fig 3, Fig 4). Classical experiments report cAMP rhythms in *Neurospora* and an ~40% increase in cAMP levels upon methylxanthine drug treatment of solid medium cultures (Scott and Solomon, 1975, *Journal of Bacteriology*, PMID: 165170). Thus, Olivares-Yañez and colleagues tested a PKA knockout strain (kinase normally activated by high cAMP levels) as well as cAMP pharmacological inhibition to more directly test the role of cAMP signaling in the circadian clock. In agreement with a recent publication (Wang et al, 2019, *Molecular Cell*, PMID: 30954403), *delta pka* does not alter circadian period (Fig 6). Pharmacological downregulation of cAMP led to just under 2 hour period lengthening (Fig 5), a more modest effect compared to the 2 - 6 hour period increases observed in the presence of methylxanthine drugs.

The present study would be of interest to the readership of *Microbiology Spectrum*, given the strong data quality and replication as well as multiple experimental approaches to investigate phosphodiesterase/cAMP/PKA signaling. However, its present form requires minor changes before publication, as detailed below.

Minor Comments (approximately in order of the main text):

Line 62: could include a more modern reference to circadian IDPs, such as Pelham JF et al 2020 PMID: 33176800 and/or Pelham JF et al 2021 bioRxiv <https://www.biorxiv.org/content/10.1101/2021.11.20.469315v1>

Line 69: should include a citation to either Sancar G et al 2011 PMID: 22152473 or Sancar G et al 2012 PMID: 23124067 to include CSP-1 in the list of TFs that can affect *Neurospora* period length

Line 101: provide the corresponding NCU numbers for all genes listed

Line 101: FGSC11513 is a heterokaryon knockout of *pkac-1*. Please clarify and confirm that all *delta pkac-1* strains presented in the results were homokaryotic for *pkac-1* knockout, which is implied in Table 1, but should be made clear

Line 104: please correct "Sacarose" to the corresponding carbohydrate source used in the medium

Lines 173 - 175: it is interesting that aminophylline and caffeine have opposite period lengthening magnitudes in race tubes compared to the luciferase plate analyses, despite very similar medium compositions. Could the authors speculate on why this might be happening? Could it have to do with nutritional compensation phenomena reported by the Dunlap lab and this group? (Emerson JM et al 2015 PMID: 26647184; Olivares-Yañez et al 2016 PMID: 27449058; Kelliher CM et al 2022 bioRxiv: <https://www.biorxiv.org/content/10.1101/2022.05.09.491261v2>)

Line 173: please clarify why Smith C et al 2010 is cited with literature on "spurious alterations in conidiation patterns"

Line 210: although not absolutely essential for publication, it would be most genetically clean to generate a conditional double knockout strain of *acon-1* and *pde-1*. This is done fairly quickly and easily in *Neurospora* as: *delta acon-1 Pqa2-pde-1* OR *Pqa2-acon-1 delta pde-1*. Such a strain would fully rule out the alternative model that residual phosphodiesterase activity in each single mutant is maintaining clock period length close to wild-type levels, instead of lengthening observed in methylxanthine drugs

Figure 5: compared to other drug treatments or mutants presented in this work, the THFA drug treatment appears to (perhaps) be delaying circadian phase as well as altering period length. Does the Biodare 2.0 platform also provide a phase shift estimate, and is the phase different between controls and THFA drug treatments?

Future work and/or improved mechanistic understanding of methylxanthine drug action on the circadian clock in *Neurospora*: while surprising that methylxanthine drugs do not act through the phosphodiesterase/cAMP/PKA signaling pathway, it is a bit disappointing that no clear mechanism or clear avenue for follow-up study emerges from this work. Many other kinases in *Neurospora* have been shown to lengthen period: *prd-3* (Mehra A et al 2009 PMID: 19450520), *prd-2* and its effect on CKI levels (Kelliher CM et al 2020 PMID: 33295874), CKI kinase inhibition with PF670462 (Hu Y et al 2021 PMID: 34182774), etc. Is there a genetic interaction between any known alleles that lengthen period and caffeine and/or aminophylline that may hint at drug mechanism?

Reviewer #2 (Comments for the Author):

This study uses a pharmacological approach to test the effects of chemicals (methylxanthines) predicted to influence cAMP levels on the period length of the circadian rhythm in *Neurospora crassa*. The authors conclude that methylxanthines affect period length independently of cAMP-phosphodiesterases. However, a major weakness of this study is that the authors did not check cAMP levels after treatment with any of the chemicals or in any mutant background.

Furthermore, as the authors note, there are 2 *pkac* catalytic subunit genes in *N. crassa* and many other fungi (*baker's yeast* has 3 genes). A double mutant should have been constructed to rule out the effect of residual activity. This is also true for the *pde*

genes-it appears that only single mutants were tested. I was surprised that the authors did not test the adenylyl cyclase mutant cr-1, which has been shown in several laboratories to completely lack detectable cAMP.

Staff Comments:

Preparing Revision Guidelines

Please return the manuscript within 60 days; if you cannot complete the modification within this time period, please contact me. If you do not wish to modify the manuscript and prefer to submit it to another journal, please notify me of your decision immediately so that the manuscript may be formally withdrawn from consideration by Microbiology Spectrum.

We thank the reviewers for valuable insights and suggestions.

We provide a point-by-point response below (*in blue*). The line numbers of the modified text are provided relative to the new MS file.

We also included some minor changes that add clarity to the work.

Reviewer #1

Line 62: could include a more modern reference to circadian IDPs, such as Pelham JF et al 2020 PMID: 33176800 and/or Pelham JF et al 2021 bioRxiv <https://www.biorxiv.org/content/10.1101/2021.11.20.469315v1>.

The suggested reference has been included (line 62)

Line 69: should include a citation to either Sancar G et al 2011 PMID: 22152473 or Sancar G et al 2012 PMID: 23124067 to include CSP-1 in the list of TFs that can affect *Neurospora* period length.

Indeed, thanks for bringing that point. Ref Sancar et al., 2011 has been now added (line 69).

Line 101: provide the corresponding NCU numbers for all genes listed.

NCUs have been added (lines 101-102).

Line 101: FGSC11513 is a heterokaryon knockout of *pkac-1*. Please clarify and confirm that all delta *pkac-1* strains presented in the results were homokaryotic for *pkac-1* knockout, which is implied in Table 1, but should be made clear.

*This has been clarified, as all *pkac-1* reporter-strains utilized in the work are homokaryons (in contrast to FGSC11513 which is indeed a heterokaryon for *pkac-1*). Moreover, Figure S1 has been added which includes the genotyping by PCR of $\Delta pkac1$, of $\Delta pkac2$ and the $\Delta pkac1$, $\Delta pkac2$ double mutant. This is also described in methods (lines 110-114). Importantly, new experiments with the double mutant have been included (Figure S10 lines 396-399), showing that rhythms remain normal in the total absence of PKA activity (see response to reviewer #2).*

Line 104: please correct "Sacarose" to the corresponding carbohydrate source used in the medium

This has been corrected (line 105).

Lines 173 - 175: it is interesting that aminophylline and caffeine have opposite period lengthening magnitudes in race tubes compared to the luciferase plate analyses, despite very similar medium compositions. Could the authors speculate on why this might be happening? Could it have to do with nutritional compensation phenomena reported by the Dunlap lab and this group? (Emerson JM et al 2015 PMID: 26647184; Olivares-Yañez et al 2016 PMID: 27449058; Kelliher CM et al 2022 bioRxiv: <https://www.biorxiv.org/content/10.1101/2022.05.09.491261v2>)

Thanks for bringing that up. This has been discussed now in lines 195-202:

"The difference in period observed in race tubes and when monitoring luciferase activity in 96-well plates, could be a consequence of how the drug may be assimilated by the actively growing fungus (race tubes) versus a more stationary state (wells). Notably, a similar phenomenon of different periods when comparing race-tubes and 96-well plates, has been described when studying circadian metabolic compensation in this fungus (9,33,34). In addition, Aminophylline is composed of two methylxanthine molecules, which, in a stationary culture may be constantly metabolized, generating higher local methylxanthine units than caffeine"

Line 173: please clarify why Smith C et al 2010 is cited with literature on "spurious alterations in conidiation patterns".

*The word spurious has been removed (as it mainly referred to banding in the *col-1* mutant), and the idea has been further elaborated (Lines 189-190).*

*"...and not just due to alterations in conidiation patterns, as the case of *col-1* where conidial bands respond to metabolic inputs (and fail to be clock regulated), or $\Delta adv-1$ where rhythmic conidiation is absent but FRQ levels oscillate as in WT, among other examples (27,30–32)"*

Line 210: although not absolutely essential for publication, it would be most genetically clean to generate a conditional double knockout strain of *acon-1* and *pde-1*. This is done fairly quickly and easily in *Neurospora* as: delta *acon-1* Pqa2-*pde-1* OR Pqa2-*acon-1* delta *pde-1*. Such a strain would fully rule out the alternative model that residual phosphodiesterase activity in each single mutant is maintaining clock period length close to wild-type levels, instead of lengthening observed in methylxanthine drugs

*This is an interesting point. The logic option is to transform $\Delta pde-1$ (as $\Delta pde-2/acon-2$ is hard to transform since it does not sporulate). Following this suggestion, we tried generating a $\Delta pde-1$ *qa-2* *acon-2*. While we observed induction by QA of the *acon-2* transcript, there was still measurable amounts of the mRNA in the absence of induction (see accompanying figure Rev#1). Race tube or slant data confirmed an "almost" *acon-2* phenotype in the absence of QA, and a fully WT appearance when the inducer was added. When it comes to clock behavior the engineered strain shows similar traces*

in the absence or presence of QA. Nevertheless, as inferred from the qPCR data, this strain is not a fully pde double KO, since expression of acon-2 is still perceivable in the absence of QA, albeit it clearly has altered PDE levels. All and all, it further validates our observations and conclusion. Yet, since it is not a bone fide double KO, we prefer not to include the data in the MS, as it may be compelling for Neurospora aficionados, but potentially confusing to other readers. However, we have incorporated those results as a Figure at the end of this letter.

Figure 5: compared to other drug treatments or mutants presented in this work, the THFA drug treatment appears to (perhaps) be delaying circadian phase as well as altering period length. Does the Biodare 2.0 platform also provide a phase shift estimate, and is the phase different between controls and THFA drug treatments?

Indeed, that is in part correct. We calculate circadian phase with Biodare 2.0 and the phase delay is mainly related to the period change. When we estimated phase considering Circadian Time (CT), we even observed that the treatment with 0.5mM and 1mM of THFA generated phase advances. However, the phase estimation shows a large deviation between the replicates, so differences are not significant (For frq^{luc} reporter Phase \pm SD. Control: 21.76 ± 1.19 ; THFA 0.3mM: 22.42 ± 1.18 ; THFA 0.5mM: 21.41 ± 1.18 ; THFA 1mM: 21.12 ± 0.85). This is now mentioned in line 328-331:

"In addition, a small change of phases is observed with the THFA treatment. Nevertheless, it seems to be related to the increase of period length, since when we estimated the phase using Circadian Time (CT), the values are very similar to the control."

Future work and/or improved mechanistic understanding of methylxanthine drug action on the circadian clock in Neurospora: while surprising that methylxanthine drugs do not act through the phosphodiesterase/cAMP/PKA signaling pathway, it is a bit disappointing that no clear mechanism or clear avenue for follow-up study emerges from this work. Many other kinases in Neurospora have been shown to lengthen period: prd-3 (Mehra A et al 2009 PMID: 19450520), prd-2 and its effect on CKI levels (Kelliher CM et al 2020 PMID: 33295874), CKI kinase inhibition with PF670462 (Hu Y et al 2021 PMID: 34182774), etc. Is there a genetic interaction between any known alleles that lengthen period and caffeine and/or aminophylline that may hint at drug mechanism?

We share the frustration expressed by the reviewer. We would have been thrilled to nail down the underlying mechanism, but we ended up focusing on obtaining compelling evidence that 1) the effect did not involve the cAMP-PKA axis and 2) confirming that other ways of altering such axis had little effect on the clock. Nevertheless, we surmise that it was relevant to report all of our findings now, as it may help interpreting other data, and may lead other colleagues to revisit aspects of the Neurospora clock and, particularly other clock models, especially when it comes to Caffeine effects.

Reviewer #2 (Comments for the Author):

This study uses a pharmacological approach to test the effects of chemicals (methylxanthines) predicted to influence cAMP levels on the period length of the circadian rhythm in Neurospora crassa. The authors conclude that methylxanthines affect period length independently of cAMP-phosphodiesterases. However, a major weakness of this study is that the authors did not check cAMP levels after treatment with any of the chemicals or in any mutant background.

Thanks for bringing up this point. We were basing our conclusions regarding cAMP on vast literature on the matter, both in Neurospora and other systems. Nevertheless, we agree that it was important to validate our interpretation of the data. Therefore, we measured cAMP levels in WT, in the absence or presence of Caffeine and 8-Br-cAMP, and also in Δ acon-2 and the cr-1 (Adenylate cyclase mutant) strain, as they relevant strains/conditions. As expected, we observed cAMP levels that reflected textbook trends in the mutants, WT and drug treated strains. This is included now in methods (lines 161-169) in Supplementary Figure 9 and commented in lines 339-352:

"Still, to make sure that intracellular cAMP levels were behaving as expected, we quantified cAMP levels by an ELISA assay (Figure S9). As control we measured the levels of cAMP in the mutant strain cr-1. As previously described, we confirmed very low levels of cAMP in this mutant (44). After the treatment with caffeine (3mM) an increase in cAMP levels is observed in the WT strain, while higher levels of cAMP are observed in Δ acon-2 compared to WT, as expected considering that in the absence of the major phosphodiesterase, cAMP degradation is impaired. Importantly, our quantification also indicated that the addition of caffeine to Δ acon-2 does not further increase cAMP levels, supporting the notion that caffeine is indeed able to increase cAMP levels by inhibiting phosphodiesterase activity, and that most of the PDE activity is due to ACON-2. Thus, the fact that caffeine in Δ acon-2 does not further increase cAMP content, but still has an impact in period length, further supports our interpretation of a cAMP-independent mechanism (Figure 2). Along the same lines, we observed a marked increase of intracellular cAMP levels when WT was grown with the cAMP analog 8-Br-cAMP (Figure S9), which emphasizes that this analog is capable of permeating the Neurospora cell membrane and, that despite its high intracellular levels, period length is not affected (Figure S7)."

Furthermore, as the authors note, there are 2 *pkac* catalytic subunit genes in *N. crassa* and many other fungi (baker's yeast has 3 genes). A double mutant should have been constructed to rule out the effect of residual activity. This is also true for the *pde* genes-it appears that only single mutants were tested. I was surprised that the authors did not test the adenylyl cyclase mutant *cr-1*, which has been shown in several laboratories to completely lack detectable cAMP.

*Following these great suggestions, the double mutant *pkac1/pkac2* was generated, genotyped (Figure S1) and evaluated regarding its clock (Figure S10). The result further confirmed our initial observation: that in the absence of PKAC activity the clock remains functional, and caffeine still exerts an effect.*

*A double *pde1/acon2* was impossible to obtain through a sexual cross (we surmise that PDE may play a critical role in part of the sexual cycle). As an alternative, we constructed, an *acon-2* tunable strain, which provided confirmatory data, but we preferred not to include those new data as we don't consider the experiment genetically clean (see response to Rev #1)*

*Finally, *cr-1* was tested and included in Figure S8, showing normal rhythms, and period lengthening in response to Caffeine. This result, in addition to *cr-1* low cAMP levels (as we also quantified them in Figure S9), provided another strong line of evidence in accordance with our interpretations on cAMP-clock function in *Neurospora*. Thanks for the suggestion!*

All these results are mentioned in lines 332-338 and, 394-399.

*"Importantly, the *Neurospora* mutant named *cr-1* exhibits compromised AC activity (44,47) leading to extremely low cAMP (48). Therefore, we evaluated the state of the oscillator in this mutant, observing that despite its very low levels of intracellular cAMP, circadian rhythms still take place (Figure S8A). Noteworthy, the expression of the output clock reporter *con-10luc* is clearly altered in this strain (Figure S8B), which is to be expected since *con-10* is a gene downstream of the PKAC-cAMP axis (49). In agreement with our prior conclusions, the addition of caffeine to *cr-1* still led to an increase in period (Figure S8C)."*

*"In addition, to rule out any role of PKA activity on the clockworks, such as that the rhythms observed in Δ *pkac-1* might be fueled by residual activity from PKAC-2, a double mutant of both kinases was obtained through sexual crosses. Evaluation of the Δ *pkac-1/pkac-2* mutants confirmed circadian rhythms similar to Δ *pkac1*, and with a period close to a WT strain, clearly establishing that PKA activity has no perceptible effect on the clockworks, at least under standard growth conditions (Figure S10B)".*

*Finally, and due to the clear results provided by the *cr-1* mutant, we included a "cautionary note" on our interpretation of the THFA data. Although, THFA has (at high concentrations), a period lengthening effect, it is puzzling that *cr-1* (which has diminished AC activity, as it should also occur by pharmacological inhibition of AC by this drug) shows normal period. Therefore, we have added the following phrase (lines 359-362):*

"On the other hand, when we genetically compromised AC activity (leading to low cAMP) period was unaffected, but when we pharmacologically blocked AC activity, an increase in period length was observed (Figure 5), which may argue of unforeseen aspects of AC on the clockworks, or alternatively, might be indicative of off-targets effects of THFA."

And line 435-438

*"Moreover, the only condition that, in our hands, led to a period change (increase) when tampering with this axis, was when utilizing the AC inhibitor THFA, yet such period outcome was not present in the AC mutant *cr-1*, result that could indicate off-targets effects of this drug, besides its known effects on AC."*

acon-2 locus

Lanes
 1: WT
 2: $\Delta pde1$
 3: $\Delta pde1 qa_{prom}::acon2$

Lanes
 1: WT
 2: $\Delta pde1$
 3: $\Delta pde1 qa_{prom}::acon2$

To confirm that the transformants had the allelic replacement of the *acon-2* promoter, the strains were analyzed by PCR. A 423 bp fragment corresponding to the *qa-2* promoter plus the beginning of the *acon-2* ORF was amplified (CO_02/CO_01). This fragment is present only in the strain with the inducible promoter (lane 3). A 1000bp genomic DNA (*rco-1* locus) was amplified as a control of the multiplex PCR reaction.

To confirm the correct insertion of the construct and that the strains are homokaryons for the modification of interest, we PCR amplified a fragment of the endogenous promoter of *acon-2* with part of the ORF (CO_05/CO_01). In WT strain this fragment has a 210bp size. In the inducible strain, the endogenous promoter has been displaced by the construct (*bar* cassette + *qa-2* promoter), so the size is larger (aprox 2000bp). As it can be seen in the gel, amplification of the small fragment is observed only in the WT strain and the $\Delta pde1$ mutant (lines 1 & 2), but not in the inducible strain, indicating that there is no WT locus present (line 3). In the *acon-2* inducible strain, only the ~2 kb band is observed.

We evaluated *acon-2* expression by adding QA to the culture medium. The strains were grown for 48 hours in LNNCCD medium with and without QA (10 mM). The expression of *acon-2* was evaluated by RT-PCR. As observed in the gel, the addition of QA increases the levels of *acon-2* transcript, however, this transcript can also be observed, to a lesser degree, in the absence of QA

We subsequently performed a qPCR in order to estimate more accurately *acon-2* mRNA levels. Expression levels were expressed relative to the control (WT strain without QA = 1). We can see that in the inducible strain *acon-2* levels are lower than in WT, but are still considerable, suggesting some leaky expression in the absence of QA. The addition of the inducer augments these levels approximately 4-fold (0.5 to 2). In the WT strain the addition of QA does not substantially modify *acon-2* expression. On the other hand, we found that in the *pde1* mutant, *acon-2* levels were elevated relative to WT, which may be part of a compensatory mechanism in the absence of *pde-1*.

Amplicon	Primer		Expected size (bp)
$QAprom$ - acon-2	CO_01	Ggatgtcctccatggcaact	423
	CO_02	tcttcaccactccgtacgtt	
acon2prom - acon2ORF	CO_05	TTGTCTTGGCCTATTGCTGG	210 (WT)
	CO_01	Ggatgtcctccatggcaact	1931 ($QAprom$)
qPCR acon-2	CO_03	CTCCGTTGACGAAATGGAAC	82
	CO_04	GTTTTCCATCGTCGATCCTG	
DNA control	oL808	ccttcaatatcatcttctgtcgtctagAAAACACCAGCGGAGTTGCAA	1.000
	oL809	gtaacgccagggtttccagtcacgacgAGAAAAAGACAAAGGTCAAG	

The *Δacon-2* strain shows a clear developmental phenotype, being unable to generate spore (conidial) bands. Since in the *qa_{prom}::acon-2* strain the expression of *acon-2* is dependent on QA, we evaluated the strain in race tubes, hoping that it would have an *acon-2*-like phenotype, which should be reversed by adding QA. *Δacon-2* and *Δpde-1*, *qa_{prom}::acon-2* strains were grown on race tubes with and without QA in a 12:12 LD photocycle. As can be seen in the upper panel, the *Δacon-2* strain does not generate bands of conidia in response to light, a phenomenon that is maintained even in QA media. In contrast, the *Δpde-1*, *qa_{prom}::acon-2*, which we expected to have a phenotype quite similar to the *Δacon-2* mutant, generates some aerial hyphae and conidia. Importantly, the addition of QA to the medium maximizes conidial banding making it look like a WT strain.

Strain	Condition	Period
WT	-QA	22.58 ± 0.14
QAprom::acon-2	-QA	22.16 ± 0.12
QAprom::acon-2	+QA	22.22 ± 0.13

The circadian period of strains with inducible *acon-2* expression was evaluated. We first evaluated what happened with expression of the *frqC-box-luc* reporter without the QA inducer. As can be seen on the top left, the central oscillator shows expression levels very similar to that observed in a WT strain with a rather unaltered period. With the addition of QA we observed an increase in signal intensity, but with period being unchanged.

May 22, 2023

Dr. Luis F Larrondo
Pontificia Universidad Catolica de Chile
Genetica Molecular y Microbiologia
Alameda 340
Alameda 340
Santiago, RM
Chile

Re: Spectrum03727-22R1 (Methylxanthines modulate circadian period length independently of the action of phosphodiesterase)

Dear Dr. Luis F Larrondo:

Thank you for your detailed revised manuscript, your manuscript has been accepted - congratulations! I am forwarding it to the ASM Journals Department for publication. You will be notified when your proofs are ready to be viewed.

Sincerely,

Rebecca Shapiro
Editor, Microbiology Spectrum
